# A Scoping Review on How to Make Hospitals Health Literate Healthcare Organizations

**DOI:** 10.3390/ijerph17031036

**Published:** 2020-02-06

**Authors:** Patrizio Zanobini, Chiara Lorini, Alberto Baldasseroni, Claudia Dellisanti, Guglielmo Bonaccorsi

**Affiliations:** 1Department of Health Sciences, University of Florence, Viale GB Morgagni 48, 50134 Florence, Italy; chiara.lorini@unifi.it (C.L.); guglielmo.bonaccorsi@unifi.it (G.B.); 2Tuscany Regional Centre for Occupational Injuries and Diseases (CeRIMP), Central Tuscany LHU, Via di San Salvi, 12, 50135 Florence, Italy; baldasse1955@gmail.com; 3Department of Epidemiology, Regional Health Agency of Tuscany, Via Pietro Dazzi, 1, 50141 Florence, Italy; claudiadellisanti70@libero.it

**Keywords:** health literacy, health literate healthcare organizations, hospital, health equity, logical framework

## Abstract

The concept of health literacy is increasingly being recognised as not just an individual trait, but also as a characteristic related to families, communities, and organisations providing health and social services. The aim of this study is to identify and describe, through a scoping review approach, the characteristics and the interventions that make a hospital a health literate health care organisation (HLHO), in order to develop an integrated conceptual model. We followed Arksey and O’Malley’s five-stage scoping review framework, refined with the Joanna Briggs Institute methodology, to identify the research questions, identify relevant studies, select studies, chart the data, and collate and summarize the data. Of the 1532 titles and abstracts screened, 106 were included. Few studies have explored the effect of environmental support on health professionals, and few outcomes related to staff satisfaction/perception of helpfulness have been reported. The most common types of interventions and outcomes were related to the patients. The logical framework developed can be an effective tool to define and understand priorities and related consequences, thereby helping researchers and policymakers to have a wider vision and a more homogeneous approach to health literacy and its use and promotion in healthcare organizations.

## 1. Introduction

Health literacy (HL) is a multidimensional concept that has been developed since the 1970s [1], moving from an individual to a public health perspective [2]. In one of its definitions, HL is described as the degree to which individuals can obtain, process, and understand the basic health information and services they need to make appropriate health decisions [3]. Thus, given that health has biological, psychological, and social determinants, HL is increasingly being recognised as not just an individual trait, but also as a characteristic related to families, communities, and organisations providing health and social services [4]. For this reason, attention has also shifted to the specific context in which care is provided: patients’ abilities to understand medical information and navigate care-seeking processes are in fact related to the demands that the health delivery systems place on them, and the specific organizational context in which care is provided may contribute to compensating for patients’ limited HL [5,6,7].

The concept of health literate health care organisations (HLHOs) is proposed to asses health care organization performance with patients’ HL issues. This kind of organization will make it easier for people to navigate, understand. and use information and services to take care of their health.

Brach et al. propose ten specific attributes (see Table 1) of HLHOs [8]. Specifically, the ten attributes of an HLHO have been described, as reported in Table 1.

Facing such attributes, many authors have proposed health literacy intervention toolkits for health care organizations [9,10], as well as correlated measurement tools [11,12,13]; however, to date a systematisation of the conceptual model based on the experiences described in the literature is still lacking.

The aim of this study is to identify and describe, through a scoping review approach, the characteristics and the interventions that make a hospital an HLHO, according to the definition of Brach et al. [8], in order to develop an integrated conceptual model capturing the most comprehensive framework of HLHOs.

The research has been restricted to hospitals instead of examining all health care settings, due to the low variability of this type of setting between different countries.

## 2. Materials and Methods

We followed Arksey and O’Malley’s five-stage scoping review framework [14], refined with the Joanna Briggs Institute methodology [15], in order to (1) identify the research questions, (2) identify relevant studies, (3) select studies, (4) chart the data, and (5) collate and summarize the data.

### 2.1. Identifying the Research Questions

The first step in the process of conducting a scoping literature review is to determine the research questions to be addressed in the study. The research question addressed in this review was based on the PCC (population–concept–context) model of the Joanna Briggs Institute [15]: “What interventions and what outcomes are pursued in health literate hospitals?” where HLHO is defined as described in the introduction [8].

### 2.2. Identifying Relevant Studies

The search strategy on Pubmed and Web of Science was built by selecting groups of keywords for each part of the PCC. Each group was combined to others through the Boolean operator AND (Table 2).

Database searches were also conducted in other four adjunctive databases: Cinahl, Scopus, Psycinfo, and Sociological Abstract. The last search was completed on 1 July 2019. No date limits were applied.

### 2.3. Selecting Articles

Two reviewers performed the data extraction and appraisal independently, with an a priori study protocol. The study protocol included the following requirements:

Only primary studies, systematic reviews, and meta-analyses were considered.

Studies should be focused on hospitals.

They must describe any intervention and outcome that concerns one or more of the ten attributes of HLHOs, as follows:Attribute 1 (leadership): studies whose aim was to involve leadership or to assess the effect of leadership involvement;Attribute 2 (planning): interventions whose aim was to introduce or test the effect of planning activities related to health literacy. This also includes every intervention aimed at developing or using tools/instruments for assessing organizational health literacy;Attribute 3 (workforce): every intervention could evaluate the impact of health literacy training on healthcare workers (HCWs), or whose aim was to develop, change, or adopt health literacy training;Attribute 4 (population): studies whose aim was to include the population in the design, implementation, and evaluation of health information and services, or every study assessing the effects of population engagement;Attribute 5 (meets the needs of the population): studies whose purpose was to assess the effect of interventions that meet “the needs of populations with a range of health literacy skills while avoiding stigmatization”;Attribute 6 (communication): studies whose aim was to implement a communication technique or that assess the effect of implementing communication techniques;Attribute 7 (navigation): studies whose aim was to implement or evaluate the impact of interventions to provide easy access to health information, both in the physical and electronic environment;Attribute 8 (contents easy to understand): every study whose aim was to assess the suitability of materials for their target audience or the impact of the development/use of suitable material;Attribute 9 (high-risk situations): every study where the intervention or outcomes were related to high-risk situations, such as informed consent for surgery, administration of medicines, advanced directives, and transitions in care, such as discharge from the hospital.Attribute 10 (payment transparency): studies whose aim was to clarify communications about health service costs to patients or evaluate the impact of interventions that make communications about insurance coverage and costs more transparent.

All electronic database search results were combined in Endnote, and duplicate records were removed. The Preferred Reporting Items for Systematic Reviews and Meta-Analyses (PRISMA) flow diagram guidance was used to display studies that were identified by the database search and met the inclusion and exclusion criteria (Figure 1).

In accordance with the standard approach to conducting scoping reviews, a quality appraisal was not performed.

### 2.4. Charting the Data

To answer the research questions, we created a data charting form in Excel with the following elements: authors, year of publication, country of the study, study design, sample characteristics, aim of the study, conclusions, related HLHO attributes, interventions, and outcome measures (Appendix).

### 2.5. Collating, Summarising, and Reporting the Results

We used information from the data charting form to summarize the overall number of studies, years of publication, countries where studies were conducted, and the focus and purpose of the studies.

The charted interventions were grouped into three categories, and each category was divided into two subcategories:Support for patients: every intervention was designed to help patients access and use health information better. Patients may receive support directly by health professionals (staff support) or by material, electronic tools, conditions, and objects belonging to the hospital’s structure (environmental support);Support for staff: interventions aimed at facilitating health professionals in helping patients. This can be achieved by health literacy training, or by tools/technologies/environments that improve the healthcare worker–patient relationship.Support for governance: interventions designed to better evaluate and manage system efforts in becoming an HLHO. This includes all the interventions aimed at developing tools/instruments for assessing organizational health literacy, as well as quality improvement actions related to health literacy: establishing measures, setting aims, specific assessment analysis, forming teams, communicating awareness, developing health literacy improvement plan, testing changes, and tracking progress.

Outcomes were also grouped into categories and subcategories:Patient outcomes: divided into changes in knowledge/skills/behaviors, perception of intervention satisfaction, and patient health outcomes.Staff outcomes: including perception of intervention satisfaction and changes in knowledge/skills/behaviors.System outcomes: including changes in the scores for tools that assess organizational health literacy, the quality improvements perceived/obtained, measures of validation for the tools developed, and costs.

The research group, with the contribution of a sociologist, developed a theoretical logical framework for a generic healthcare organization and combined it with interventions and outcomes to better synthetize the data. A logical framework is a diagram mapping out a chain of hypothesized causal relationships and providing a structure to describe the interventions that are available to reach specified public health goals [16]. The purpose of a logical framework is organizing, grouping, and selecting the interventions for the health issues under consideration, and for choosing the outcomes used to define the success of each intervention [17].

## 3. Results

Figure 1 shows the article selection.

Of the 1532 titles and abstracts screened, 106 were included, of which 97 were primary studies and 9 were systematic reviews. Among the primary studies, 24 were randomized controlled trials, 42 were quasi-experimental studies, 19 were descriptive studies, and 12 were validation studies. The majority (70%) of the selected primary studies were performed in the United States, followed by Europe (13%), Australia (6%), Canada (3%), and others (6%).

In Table 3 and Table 4, respectively, the interventions and outcomes divided into subcategories are summarized.

### 3.1. Interventions (See Table 3 for References)

The majority of studies investigate the effects of interventions that support patients. A total of 66 [18,19,20,21,22,23,24,25,26,27,28,29,30,31,32,33,34,35,36,37,38,39,40,41,42,43,44,45,46,47,48,49,50,51,52,53,54,55,56,57,58,59,60,61,62,63,64,65,66,67,68,69,70,71,72,73,74,75,76,77,78,79,80,81,82,83] have investigated environmental support. This includes both material support (informative brochures, flyers, or pamphlets) and digital technology (software/apps) for patient education or to help patients to better access or manage their health information. Thirty-five studies investigated the effects of staff interventions in helping patients by describing interventions aimed at educating or helping patients in their healthcare pathways, such as during medical reconciliations and follow-ups. Only 15 studies examined the effect of interventions targeting hospital staff. Most of them (12) were related to health literacy training programs to improve staff communication skills, and six examined interventions to help doctors create easily understandable material, such as templates for discharge instructions.

Fifteen studies examined interventions to support hospital governance: 10 illustrate the development and use of instruments or tools for assessing organizational health literacy, while 7 sought to evaluate quality improvements such as health literacy interventions and activities.

### 3.2. Outputs/Outcomes (See Table 4 for References)

The most common outcome was a change in knowledge/skill/behaviour of patients (57 studies); twenty-three studies used subjective outcomes, such as perceptions of satisfactions and helpfulness, and 22 studies reported patient health outcomes. Only 15 studies used staff outcomes: 12 for changes in knowledge/skill/behaviour, and 6 for subjective perceptions. Fifteen studies have analysed system outcomes: 11 of these have utilized scores of tools for conducting organizational assessments, 6 have measured quality improvement changes, 8 have measured the validity of particular tools developed by the organization, and 3 have evaluated the costs related to the interventions.

### 3.3. Attributes (See Table 5 for References)

The most common attribute investigated was the eighth (with 67 studies, followed by attribute 9 with 53 studies and 7 with 36 studies]. No study was found to be related to attribute 10. Attribute 5 is very generic, so it can be included in any of the remaining nine attributes. This will be further explained in the discussion section. Only 18 studies (see Appendix A for references) analysed a single attribute.

### 3.4. Logical Framework (Figure 2)

The logical framework developed depicts the relationship between the HLHO and its intended effects on patient health outcomes. It is composed of the determinants of an HLHO (health literacy definition, reference population, and features of the organization) and phases (governance, staff, and environment) that identify the ways in which HLHOs interact with patients. At the end of the logical framework, patient health outcomes are shown. However, this was not meant to be a definitive guide to the relationship between these components, because many of these relationships have not been explicitly tested. Each phase has its own interventions and proximal outcomes (outputs) that define success for each intervention. Interventions and outputs, obtained by analysing the results of our literature search, were organized in a matrix linked to the phases. Governance interventions were applied only to the governance phase, while interventions of support staff and patients were applied to both the staff and environmental phases.

## 4. Discussion

In the years following its publication, the “Ten Attributes of Health Literate Health Care Organizations” has been used as both an assessment tool and a guidebook for building health literate organisations [10,123]. However, the 10 attributes have been criticized for being developed inductively and for lacking theoretical backing [124]. With our work, we have tried to overcome this issue by searching and analyzing the literature and building a logical framework to support the 10 attributes in defining an HLHO.

Most literature has investigated attributes 8 and 9, which combined together are related to the use of educational material and are strictly related to the environmental phase of our logical framework. No study was found to be related to the 10th attribute. This could be because patients do not have to pay for services in every health care system. However, in our review, the majority of studies were set in the United States. Clear communication of costs for patients is not considered to be an issue related to health literacy. In addition, our work evidenced that the majority of interventions belonged to two or more attributes. The cause of this is to be found in some redundancies related to the 10 attributes. The fifth attribute is a perfect example of how difficult it is to define the limits of the domains for each attribute. The description says that an HLHO “meets the needs of populations with a range of health literacy skills while avoiding stigmatization”. If the focus is meeting the needs of a population, then every intervention found in our review could belong to this attribute; on the contrary, when interventions focus on just avoiding stigmatisations, then no study could be related to this attribute. While this redundancy can be useful for broadly describing every aspect of an HLHO, it can generate confusion at a decision-making level. Thanks to our logical framework, it should be easier to identify and to determine to which area every type of intervention belongs. For instance, as described above, most literature reports interventions and outcomes related to attributes 8 and 9, but this does not give enough information about the context they refer to. Analyzing our results using our logical framework, we can clearly see that the vast majority of research has investigated the role of interventions to support patients, while focusing on environmental content that is easy to understand and act on. For this reason, the most common types of outcomes reported were related to the patients—in particular, changes in knowledge, skill, and behaviour. However, this process includes, for the most part, materials related to patient education, and no study was found on navigation issues in the physical environment of a hospital, even though navigation was first raised as a health literacy problem out of concern for the complexity of health care facilities and their poor signage. There are many interventions, like using color coded pathways, standardizing plain language directions, having volunteer escorts, and posting directions in commonly used languages and navigation apps [125], but we did not find any studies that evaluated their effects.

In addition, a reasonable number of patient health outcomes were investigated, but they were all related to interventions to support patients. No study related to staff or governance intervention reported any kind of patient health outcomes.

Very few studies [50,64,67,68,76,110] have explored the effect of environmental support on health professionals, and few outcomes related to staff satisfaction/perception of helpfulness have been reported [23,100,101,102,103,104]. At the same time, studies examining interventions to support the governance of the organization, despite receiving more attention, often had methodological limitations, due to their weak study designs. As such, the generalizability of the findings from these studies was limited. For example, some of them were on using organizational health literacy assessment tools. While these tools were all pilot-tested for overall usability, none of them were clearly demonstrated to be reliable to measure improvement. In the literature, it is clear that limited health literacy is a significant factor associated with increased healthcare utilization and costs [126,127], and that meeting the needs of people with limited health literacy could produce savings of approximately 8% of the total costs for this population [127]. However, only three studies [40,80,98] reported “costs” as an outcome. Health organizations need resources and strategies to save staff time and costs [128]. It would be desirable to justify health literate interventions by linking them to saving staff time or reducing costs to convince more health organizations (including business-driven health organizations) to transform themselves to meet health literacy goals.

Our work has some limitations. First, we only take hospitals into account, and due to the characteristics of a scoping review, we did not evaluate the quality of studies. Consequently, this review cannot report the best intervention for a health organization wishing to become health literate. Our logical framework shows every group of interventions and related consequences reported by the literature so far. Hopefully, this would help researchers and policymakers to move beyond the single-intervention-based improvement mindset, and to implement groups of interventions for each area of the organization, making health literacy integral to all operations. When an organization sets a goal of becoming health literate, it replaces fragmented quality improvement activities with a systematic and comprehensive approach [129].

## 5. Conclusions

This scoping review identifies and describes the characteristics and the interventions that make a hospital an HLHO. So far, in the literature, little attention has been given to the effect of environmental support on health professionals, and few outcomes related to staff satisfaction/perception of helpfulness have been reported; the most common types of interventions and outcomes reported have been related to the patients.

We also build a logical framework here to support the 10 attributes in defining an HLHO, which, despite some limitations, can be an effective tool to better define and more specifically understand priorities and related consequences, thereby helping researchers and policymakers to have a wider vision and a more homogeneous approach to health literacy and its development in healthcare organizations.

## Figures and Tables

**Figure 1 ijerph-17-01036-f001:**
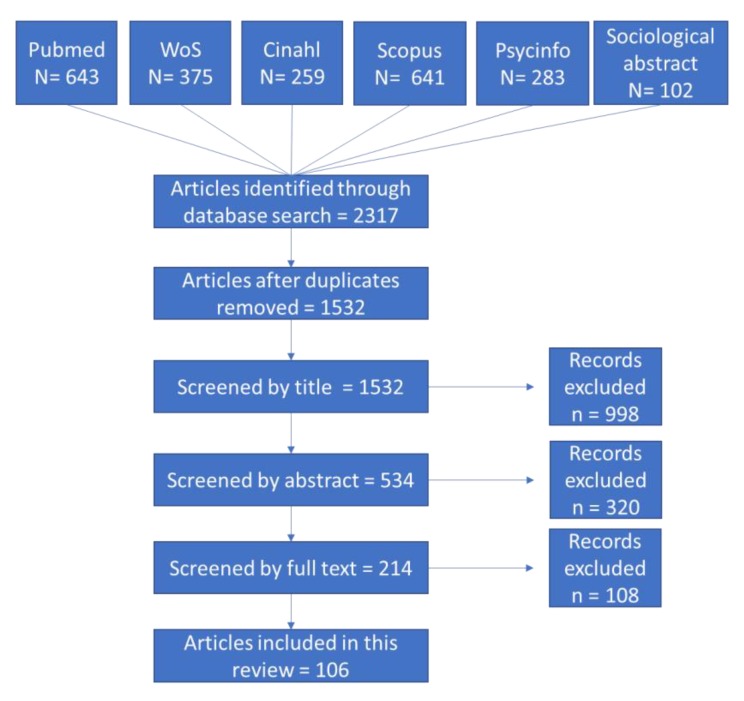
Preferred Reporting Items for Systematic Reviews and Meta-Analyses (PRISMA) flow diagram.

**Figure 2 ijerph-17-01036-f002:**
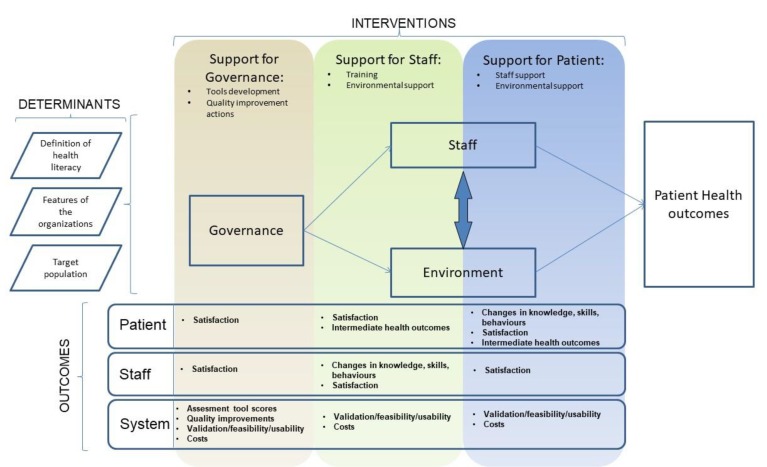
Logical framework.

**Table 1 ijerph-17-01036-t001:** Ten attributes to the health literate health care organizations (HLHOs), according to the Institute of Medicine (IOM) [8].

An HLHO
Has leadership that makes health literacy integral to its mission, structure, and operations. Integrates health literacy into planning, evaluation measures, patient safety, and quality improvement. Prepares the workforce to be health literate and monitors progress. Includes populations served in the design, implementation, and evaluation of health information and services. Meets the needs of populations with a range of health literacy skills while avoiding stigmatisation. Uses health literacy strategies in interpersonal communications and confirms understanding at all points of contact. Provides easy access to health information and services, as well as navigation assistance. Designs and distributes print, audiovisual, and social media content that is easy to understand and act on. Addresses health literacy in high-risk situations, including care transitions and communications about medicines. Communicates clearly what health plans cover and what individuals will have to pay for services.

**Table 2 ijerph-17-01036-t002:** Search strings.

Strings	Database
((“health literacy” AND implementation) OR (“Health Literacy/nursing”[Mesh] OR “Health Literacy/organization and administration”[Mesh] OR “Health Literacy/utilization”[Mesh]) OR “health literate” OR (“health literacy” AND (organizat * OR organisat *)) AND (“hospitals”[MeSH Terms] OR “hospitals”[All Fields] OR “hospital”[All Fields] OR hospital * OR “health facility *” OR “Health Facilities”[Mesh])	Pubmed
((“health literacy” AND implementation) OR (“Health Literacy” AND nursing) OR ((“Health Literacy” OR “health literate”) AND (organizat * OR administrat * OR utilizat *))) AND (hospital * OR “health facility *”)	WoS, Cinahl, Scopus, Psycinfo
“health literacy” OR “health literate”	Sociological Abstract

**Table 3 ijerph-17-01036-t003:** Interventions (see Appendix A for more detail).

Intervention	Subcategories	Code	Ref.	No. of Studies
**Support for patient**	Environmental	1a	[18,19,20,21,22,23,24,25,26,27,28,29,30,31,32,33,34,35,36,37,38,39,40,41,42,43,44,45,46,47,48,49,50,51,52,53,54,55,56,57,58,59,60,61,62,63,64,65,66,67,68,69,70,71,72,73,74,75,76,77,78,79,80,81,82,83]	66
	Staff	1b	[19,21,24,28,29,30,34,40,41,49,51,61,63,66,67,68,75,80,82,83,84,85,86,87,88,89,90,91,92,93,94,95,96,97,98,99]	35
**Support for staff**	Training	2a	[67,80,100,101,102,103,104,105,106,107,108,109]	12
	Environmental	2b	[50,64,67,68,76,110]	6
**Support for governance**	Developing/usig tools/instruments for assessing organizational health literacy	3a	[11,108,111,112,113,114,115,116,117,118]	10
	Actions for quality improvements	3b	[50,111,116,119,120,121,122]	7

**Table 4 ijerph-17-01036-t004:** Outcomes (see Appendix A for more detail).

Target	Subcategories	Code	Ref.	No. of Studies
**Patient**	Knowledge/skills/behaviour	1a	[18,22,23,24,25,26,27,29,30,31,32,33,35,36,37,38,40,41,42,43,44,46,47,49,50,51,52,53,54,55,57,59,60,61,63,65,66,68,70,75,76,77,78,79,80,81,82,83,87,88,93,95,96,121,122]	57
	Satisfaction/acceptability/helpfulness/	1b	[20,21,22,28,30,31,47,48,54,56,59,63,67,73,79,81,84,85,94,106,107,117,120]	23
	Patient health outcomes	1c	[19,27,34,37,41,45,63,66,71,76,80,86,87,89,90,91,92,93,97,98,99]	22
**Staff**	Knowledge/skills/behaviour	2a	[38,64,67,68,101,102,103,105,108,109,110,122]	12
	Staff perception of satisfaction/helpfulness	2b	[23,100,101,102,103,104]	6
**System**	Assessment tools scores (organisation)	3a	[11,84,94,116,117,121]	11
	Quality improvements perceived/gained	3b	[108,111,116,118,119,122]	6
	Validation/feasibility/usability/	3c	[39,58,69,72,114,117]	8
	Costs	3d	[40,80,98]	3

**Table 5 ijerph-17-01036-t005:** Attributes (see Appendix A for more detail).

N	Description	No. of Studies	Ref.
1	Leadership	7	[65,105,106,111,114,115,116]
2	Planning	19	[11,69,72,73,88,94,106,107,110,111,112,113,114,115,116,117,119,121,122]
3	Workforce	13	[67,80,94,100,101,102,103,104,105,106,107,109,110]
4	Population	10	[11,28,38,42,50,56,58,59,67,118]
5	Meets the needs of the population	0–106	*
6	Communication	25	[19,36,38,41,49,51,61,63,67,78,79,80,82,89,90,93,94,96,99,100,101,102,104,107,109]
7	Navigation	36	[19,20,21,24,28,30,31,32,34,36,38,40,46,48,53,54,56,57,58,59,60,63,68,72,82,84,86,87,89,90,97,98,99,112,121,122]
8	Contents easy to understand	67	[18,19,20,22,23,24,25,26,27,28,29,30,31,32,33,35,36,37,38,39,40,41,42,43,44,45,47,49,50,51,52,53,54,55,56,57,58,60,61,62,63,64,65,66,67,69,70,72,73,74,75,76,77,78,79,80,81,82,83,90,93,110,112,118,120,121,122]
9	High-risk situations	53	[18,19,20,21,23,25,26,27,28,32,34,35,37,38,39,40,41,43,44,45,49,52,54,59,61,62,63,64,65,66,67,68,74,75,76,80,82,83,84,85,86,87,88,90,91,92,93,95,96,98,99,113,119]
10	Payment transparency	0	/

* further explained in the Discussion section.

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
