# Peer review of "A Scoping Review on How to Make Hospitals Health Literate Healthcare Organizations"

_ijerph, 2020, doi:10.3390/ijerph17031036_

Round 1

Reviewer 1 Report

There is need for significant review of the logic model and the some of the definitions and measures elucidated. Some inconsistencies are identified (see comments in the manuscript).

Long, convoluted sentences and numbers in sentences reduce readability. Some precision in language needed (e.g. "In fact" is not technically precise). Reduce and summarize the number of numbers in the narrative.

Models with confounders (Table 5, N5) should be clarified further, either by disaggregating the measures (e.g. measures of "population needs"), or by eliminating the confounding item altogether, especially if other items already contain measures defining that confounder.

Summary frequency distributions should be included in tables 3 (interventions) and 4 (Outputs) and only significant findings reported in the narrative.

Detailed comments in manuscript.

Author Response

1- Title

R- Please review and revise the title. According to the JBI Manual (2017),  chapter 11, Scoping Review Title should not be framed as a question.

A- Changed

2- Line 28

R- Delete

A- Deleted

3- Line 43-46

R- Sentence long and convoluted. Consider: "The concept of HLHOs is proposed, to asses health care organization performance performance with pations' HL issues. This concept will make it easier for people to navigate, understand and use information and services, to take care of their health. Brach et al propose ten specific attributes (see table 1) of HLHOs[8]."

A- Changed in: “The concept of HLHOs is proposed to asses health care organization performance with patiens' HL issues. This kind of organization will make it easier for people to navigate, understand and use information and services, to take care of their health.

Brach et al propose ten specific attributes (see table 1) of HLHOs [8].

4- Line 58

R- Expression lacks scientific precision. Consider more precise terminology

A- This sentence was a bit misleading while adding little to the text. We deleted it entirely

5- Line 146

R- staff outcomes should include ease of utilization and application as well as overall paitent outcomes, reflecting the effects of changes in staff outcomes on the hospital's patient outcome objectives

A - We have considered those staff outcomes that should generate direct changes to the staff. “Ease of use” may be considered a specific quality, or characteristic of a specific tool rather than a staff outcome. In the logical framework the correlations between different groups of outcomes are shown.  

6- Line 148

R- These are process outcomes rather than governance issues. Change title or change definition

A- Changed in “system outcomes” to match the logical framework

7- Line 159

R- These results are not illustrated in any table. Table 2 appears to imply these are to be seen there. Either produce a descriptive table of these items or reframe the statement to separate the two

A- Briefly changed line 159.

8- Table 4

R- Harmonize language in this tabular output with section 2.5

A- Changed in “outcomes”

R- Summary frequencies/proportions column is needed in this table

A- Added

9- Line 170-197

R- Too many numbers in sentence. Sentence long and unwieldy

A- Long sequences of citations deleted. Sentence reframed

10- Table 5

R- This attribute should not be included if it is embedded in others. On the other hand the "needs of the population" referred to should be elucidated.

A- We added a reference in the discussion to explain more in detail our approach

11- Figure 2

R- Governance themes are not clearly reflected in "Support for Governance" Align measures with themes

A- As described in line 213, they are not governance themes, but  determinants of the HLHO. We changed the image to make it clearer.

R- In developing a standard evaluation tool for staff use, Tools Validation (including ease of use) should be under staff support

A- See section 5. Image changed according to the suggestion.

R- Should be reflected in narrative

A- Changed some words in system outcomes to better match the narrative

R- Print too small

A- Changed image dimensions

12- Line 235

R- Having identified this weakness, researchers should address it and report what was done to reduce or eliminate the weakness in methodology within the context of their logic model.

A- Sentence added - “Thanks to our logical framework it should be easier to identify and determine to which area every type of intervention belongs. For instance, as reported above, most literature reports interventions and outcomes related to attributes 8 and 9 but this does not give enough information about the context they refer to”.  

13- Line 270

R- Describe how this study logic model developed moves knowledge along this concept

A- Sentence added-  “Our logical framework shows every groups of interventions and related consequences reported by the literature so far. Hopefully this would help researchers and policymakers to move beyond the single-intervention-based improvement mindset, and to implement groups of interventions for each area of the organization, making health literacy integral to all operations.”

Reviewer 2 Report

Dear authors,

You have done a lot of work. I think your work will contribute to this area.

I will write some tiny comments to make your research better.

line 85.

The authors describe ten attributes of HLHOs. Is there any previous study written about these 10 attributes?

table 3.

It would be better to put a horizontal line tin the table to make it easier to identify the classification.

Although "staff" is written in the table, the author use the word " health professional". If there is no difference between two words, it is better to use the same word.

Since "code" is not used anywhere else in this table, I feel "code" is not necessary.

table 4.

The title of the table is "outputs", however, the word used in the main text is "outcome". 

"system" is described as "governance" in main text.

line 185-188.

I think the contents described in this part is refer to "support for governance" in table 3. However, the words are very different and may be confusing.

table 5.

It might be easy to understand, if the "description" used in table 5 is written on "ten attributes" in page 3. 

In addition, I think that "No of studies" should be in table 3 and 4.

Figure 2.

Is it possible to add the elements of "ten attributes" to the figure?

Author Response

1- line 85

R- The authors describe ten attributes of HLHOs. Is there any previous study written about these 10 attributes?

A- Brach et al. were the first to describe the ten attributes of HLHOs. Our writing probably was not clear enough so we changed line 44-47 to provide a better explanation.

2-Table 3

R- It would be better to put a horizontal line tin the table to make it easier to identify the classification.

A- Changed as suggested

R- Although "staff" is written in the table, the author use the word " health professional". If there is no difference between two words, it is better to use the same word.

A- Changed “health professionals” with staff

R- Since "code" is not used anywhere else in this table, I feel "code" is not necessary.

A- Code refers to the tables in the supplementary file that provide a detailed description of every article. We think that it is needed to clearly explain how authors grouped every intervention and outcome. For the sake of clarity we added “See supplementary file for more details” to the tables description.

R- The title of the table is "outputs", however, the word used in the main text is "outcome". 

A- Changed as suggested

R- "system" is described as "governance" in main text.

A- Changed as suggested

3- line 185-188.

R- I think the contents described in this part is refer to "support for governance" in table 3. However, the words are very different and may be confusing.

A- Added “quality improvements such as” in line 183 to better match words in table 3

4- table 5.

R- It might be easy to understand, if the "description" used in table 5 is written on "ten attributes" in page 3. 

A- Changed as suggested

R- In addition, I think that "No of studies" should be in table 3 and 4.

A- N. of studies added in table 3 and 4

5- Figure 2.

R- Is it possible to add the elements of "ten attributes" to the figure?

A- We think that, although it could be interesting to add some further details regarding the ten attributes, the logical framework might become too complicated and difficult to understand.

Round 2

Reviewer 1 Report

This article is an important addition to growing interest in hospital health literacy as a specific adjunct to care effectiveness and quality. A logic model helps organize and support hypotheses development and prediction.

Review tone of title and article body, and realign (optional).
